# Telomere Length, Mitochondrial DNA, and Micronucleus Yield in Response to Oxidative Stress in Peripheral Blood Mononuclear Cells

**DOI:** 10.3390/ijms25031428

**Published:** 2024-01-24

**Authors:** Andrea Borghini, Rudina Ndreu, Paola Canale, Jonica Campolo, Irene Marinaro, Antonella Mercuri, Stefano Turchi, Maria Grazia Andreassi

**Affiliations:** 1CNR Institute of Clinical Physiology, 56124 Pisa, Italy; rudina.ndreu@cnr.it (R.N.); paola.canale@santannapisa.it (P.C.); irene.marinaro@cnr.it (I.M.); antonella.mercuri@cnr.it (A.M.); stefano.turchi@cnr.it (S.T.); mariagrazia.andreassi@cnr.it (M.G.A.); 2Health Science Interdisciplinary Center, Sant’Anna School of Advanced Studies, 56124 Pisa, Italy; 3CNR Institute of Clinical Physiology, ASST Grande Ospedale Metropolitano Niguarda, 20142 Milan, Italy; jonica.campolo@cnr.it

**Keywords:** telomere length, chromosomal damage, micronucleus, mitochondrial DNA copy number, hydrogen peroxide, oxidative stress, peripheral blood mononuclear cells

## Abstract

Telomere shortening, chromosomal damage, and mitochondrial dysfunction are major initiators of cell aging and biomarkers of many diseases. However, the underlying correlations between nuclear and mitochondrial DNA alterations remain unclear. We investigated the relationship between telomere length (TL) and micronucleus (MN) and their association with mitochondrial DNA copy number (mtDNAcn) in peripheral blood mononuclear cells (PBMCs) in response to 100 μM and 200 μM of hydrogen peroxide (H_2_O_2_) at 44, 72, and 96 h. Significant TL shortening was observed after both doses of H_2_O_2_ and at all times (all *p* < 0.05). A concomitant increase in MN was found at 72 h (*p* < 0.01) and persisted at 96 h (*p* < 0.01). An increase in mtDNAcn (*p* = 0.04) at 200 µM of H_2_O_2_ was also found. In PBMCs treated with 200 µM H_2_O_2_, a significant inverse correlation was found between TL and MN (r = −0.76, *p* = 0.03), and mtDNA content was directly correlated with TL (r = 0.6, *p* = 0.04) and inversely related to MN (r = −0.78, *p* = 0.02). Telomere shortening is the main triggering mechanism of chromosomal damage in stimulated T lymphocytes under oxidative stress. The significant correlations between nuclear DNA damage and mtDNAcn support the notion of a telomere–mitochondria axis that might influence age-associated pathologies and be a target for the development of relevant anti-aging drugs.

## 1. Introduction

Genome instability driven by a large variety of endogenous and exogenous insults is a major factor in aging and most common non-communicable diseases, which are the leading cause of death and disability [1,2]. Indeed, a large number of correlative studies have revealed evidence for an accumulation of nuclear DNA damage in aging and chronic diseases, which can contribute to impairments in the maintenance and function of cells and tissues [1,2]. Accordingly, there is a growing interest in identifying markers of human aging to be used in clinical settings to predict disease risk and outcomes, as well as to develop molecular therapies and lifestyle interventions that slow down the aging process.

The frequency of micronuclei (MN) in peripheral blood lymphocytes is extensively used as a surrogate marker of chromosomal damage and genomic instability in different tissues [1]. MN are extra-nuclear bodies that contain damaged chromosome fragments and/or whole chromosomes that were not incorporated into the nucleus after cell division. A large number of studies have consistently shown the association between an increase in MN frequency and the risk of cancer, as well as several aging-related diseases [3].

Additionally, it is well known that telomere integrity is crucial to maintaining genome stability [4]. Telomeres are hexameric DNA repeats (TTAGGG) in association with a complex of proteins that protect the ends of the chromosomes by capping them and preventing their end-to-end fusion during cell division [5]. Conversely, telomere erosion is a hallmark of cell senescence (a quiescent, non-replicative state) that drives cell dysfunction or apoptosis. Indeed, short telomeres have been strongly implicated in the risk of developing many diseases, ranging from cancer to cardiovascular disease and neurodegeneration [5,6].

Moreover, increasing evidence points to emphasizing the significance of the intimate link between telomeres and mitochondrial metabolism in the process of cellular senescence and the onset of age-related diseases [7,8,9]. At present, the mitochondrial DNA copy number (mtDNAcn) is a promising marker of aging and mitochondrial function in clinical and population studies, as the levels of mtDNAcn are directly correlated with energy reserves, respiratory enzyme function, and mitochondrial membrane potential [10].

Interestingly, intriguing studies indicate that changes in the mtDNAcn might contribute to mitochondrial dysfunction and consequent increased ROS production in the mitochondrial matrix, which may further exacerbate nuclear DNA and telomere damage in a vicious cycle [7,9].

However, little is known about the underlying correlations between telomere length (TL), chromosomal damage, and abnormalities in the mtDNA. The present study is designed to investigate the relationship between telomere attrition and chromosomal damage evaluated via a cytokinesis-blocked micronucleus (CBMN) assay and whether nuclear DNA alterations can affect mtDNAcn in peripheral blood mononuclear cells (PBMCs) activated in vitro by phytohemagglutinin (PHA) in the presence/absence of hydrogen peroxide (H_2_O_2_) as an oxidative and genotoxic stimulus.

## 2. Results

### 2.1. Effect of Oxidative Stress on Telomere Length and Chromosomal DNA Damage

For both 100 µM and 200 µM H_2_O_2_ treatments, we observed dose-dependent increases in telomeric damage at different times (*p* < 0.05). Significant telomere shortening was observed after exposure to either dose of H_2_O_2_ (100 µM, *p* = 0.03; 200 µM, *p* = 0.005) compared to the control at 44 h (Figure 1). The telomere length reduction was about 26% and 38% for treatments with 100 µM and 200 µM H_2_O_2_, respectively. The decrease in telomere length also persisted at 72 h and 96 h after PBMC incubation (*p* < 0.05) (Figure 1).

We also performed the CBMN assay to test whether treatment with oxidative stimulus-induced chromosomal damage (Figure 2). The results revealed a dose-related increase in MN after H_2_O_2_ treatment (*p* < 0.05). A significant increase in micronuclei was found at 72 h (18 ± 2 vs. 32 ± 3 and 41.5 ± 2 for baseline, 100 µM, and 200 µM) and persisted at 96 h (19 ± 2.5 vs. 33 ± 3 and 37.5 ± 2 for baseline, 100 µM, and 200 µM, respectively).

### 2.2. Effect of Oxidative Stress on Mitochondrial DNA Copy Number

For mtDNAcn analysis, no significant difference in mtDNAcn was observed in the treated samples at 44 h and 72 h (*p* > 0.05) (Figure 3). Conversely, our results demonstrated a significant increase in mtDNAcn (559 ± 84 vs. 1090 ± 177, *p* = 0.04) after high oxidative stress exposure (200 µM H_2_O_2_) compared to the control value. A moderate trend toward significance was also observed after exposure to 100 µM H_2_O_2_ (*p* = 0.08, Figure 3).

### 2.3. Correlation between Nuclear DNA Damage and Mitochondrial DNA Copy Number

When looking for a potential association between nuclear DNA damage and mtDNA content, we found a significant inverse correlation between TL and MN frequency in PBMCs treated with 200 µM H_2_O_2_ (r = −0.76, *p* = 0.03) (Figure 4). Additionally, TL was found to be correlated with mtDNA content after high oxidative stress exposure (r = 0.6, *p* = 0.04).

After the same exposure to 200 µM H_2_O_2_, an significant inverse correlation was found between MN frequency and mtDNAcn (r = −0.78, *p* = 0.02, Figure 4).

## 3. Discussion

In the present study, we first observed a shortening in TL and a concomitant increase in chromosomal DNA damage after treatment with H_2_O_2_ in PBMCs, supporting the notion that the decreased length of telomeric DNA sequences is clearly associated with increased chromosomal damage.

It is becoming increasingly evident that the primary role of telomeres and their associated proteins is to maintain chromosome and genome stability by preventing chromosomal ends from being recognized as double-strand breaks and protecting them from end-to-end fusion and degradation [4,11]. While acting as factors to prevent loss of genetic information, telomeric DNA shortens in somatic cells with each cell division [4]. As telomere length declines with aging, cancer and other non-communicable diseases have been linked to this shortening [5,6]. Telomeric erosion is primarily due to oxidative stress that preferentially damages telomeric regions over the other genomic DNA regions, as well as inhibits telomerase activity in vitro in various cell types [12]. During oxidative stress, the accumulation of DNA damage within telomeres is enhanced by the high incidence of guanine residues in telomeric DNA sequences [13] and by a less efficient repair process compared with the rest of the genome [14].

The loss of telomeric repeat sequences or deficiencies in telomeric proteins can result in chromosome fusion and lead to chromosome instability [15]. Chromosome instability related to telomere dysfunction is mainly mediated by the formation of nuclear anomalies such as MN, which represent established markers of genotoxic events and genome instability [16,17]. MN originates from chromosome fragments or intact chromosomes that are not included in daughter nuclei during mitosis. The main reasons for their formation are a lack of functional centromere in the chromosome fragments or whole chromosomes or defects in one or more of the proteins of the mitotic system that, consequently, fail to properly segregate chromosomes [18]. Accordingly, dysfunctional telomeres can generate a wide variety of chromosome-visible alterations as MN, generating unstable karyotypes associated with human aging and cancer [15].

In our study, the analysis of TL revealed significant telomere shortening after treatment at 44 h that persisted at 72 h and 96 h. Interestingly, an increase in chromosome instability was found in the same time frame of telomere shortening, suggesting a clear correlation between these two markers of nuclear genome instability. In particular, the strict correlation with TL indicates that the analysis of MN represents a good readout for telomere defects and any resulting chromosome segregation errors.

Our results are consistent with previous observations by Coluzzi et al. [19], which reported that oxidative base damage leads to abnormal nuclear morphologies (nucleoplasmic bridges, nuclear buds, and MN frequency), and telomere dysfunction is an important contributor to this effect. Even though the authors found telomere shortening and increased abnormal nuclear morphologies 48 h after treatment in human primary fibroblasts, they observed a restoration of TL and a reduction in chromosome instability, especially nucleoplasmic bridges, at subsequent times (72 and 96 h) [19]. This is probably due to the different times of treatment with H_2_O_2_ since cells were only treated with hydrogen peroxide (100 and 200 μM) for 1 h against persistent treatment with the oxidative stimulus, as in our study.

To the best of our knowledge, this is the first study in the literature evaluating the in vitro responses to H_2_O_2_ of both telomere and chromosomal damage with changes in mtDNAcn in PBMCs. PBMCs are typically in the resting stage of the cell cycle (G0) and can be stimulated by mitogens to divide in vitro. Among the mitogens, phytohemagglutinin (PHA) stimulates the T-cell (thymus dependent) fraction of lymphocytes, while it has little or no effect on the B-cell (bone-marrow-dependent) lymphocyte fraction [20]. T-lymphocytes are of particular interest since they play an important role in the control of the immune response and against noxious agents in several diseases [21]. Additionally, lymphocytes are commonly used as surrogate cells to measure specific genetic alterations of other cells, representing a useful cell model for studying human aging and multiple diseases [22]. Thus, our findings indicated that both MN and TL measurements in PBMCs may be sensitive markers of oxidative stress, nuclear genetic damage, and cell aging, as well as indicators of the risk and progression of common aging pathologies.

Mitochondrial dysfunction is another sign of aging, and as cells have both a nuclear and a mitochondrial genome, this is also intimately tied to genomic instability. Mitochondria are both the major intracellular source and primary target of ROS [23]. According to the free radical theory, ROS causes oxidative DNA damage in both the mitochondrial and nuclear genome, which results in an accumulation of mutations that eventually lead to aging [24]. The evidence suggests that mitochondrial DNA (mtDNA) damage plays a role in contributing to mitochondrial dysfunction and subsequent reactive oxygen species (ROS) production, thereby intensifying telomere damage in a cyclic manner [7,9]. It is noteworthy, however, that mitochondria can also become dysfunctional as telomeres shorten, establishing a connection between the primary theories of cellular aging [25]. Telomere dysfunction triggers the suppression of the master regulator of mitochondrial biogenesis and function, namely peroxisome proliferator-activated receptor gamma co-activator 1α/β (*PGC-1α/β*), resulting in mitochondrial dysfunction and ROS production. Specifically, telomere attrition activates *p53* and DNA damage response pathways, which, in turn, inhibit *PGC-1α* and *PGC-1β*. Consequently, mitochondrial dysfunction driven by telomere shortening may impair oxidative phosphorylation and increase ROS generation. This, in turn, can further exacerbate mitochondrial distress and telomere shortening, creating a two-pronged vicious cycle that amplifies the dysfunctional system [26].

Each human cell contains about 100–1000 mitochondria carrying 2–10 copies of mtDNA. mtDNAcn is positively correlated with the number and size of mitochondria. Compared with nuclear DNA, mtDNA has diminished protective histones and DNA repair capacity and is, hence, particularly susceptible to ROS-induced damage. Cells challenged with ROS have been shown to synthesize more copies of their mtDNA and increase the mitochondrial abundance to compensate for damage [27].

Together with telomere loss and chromosome instability, our data revealed that the mtDNA contents of cells treated with H_2_O_2_ increased with the duration of the incubation, leading us to hypothesize a potential compensatory mechanism to oxidative stress-induced telomeric loss. However, our study observed a significant correlation between nuclear DNA damage and mtDNA copy number, supporting the notion of a telomere–mitochondria axis that directly links DNA damage, telomere shortening, and the direct effect on mitochondrial DNA content via p53 [28].

Additionally, our findings are consistent with previous studies indicating that telomeric DNA damage occurs earlier than mitochondrial failure [29]. T cells treated with KML001 (sodium meta-arsenite) showed telomere dysfunction that, in turn, activated the *p53* signaling pathway. *p53* regulation can influence the expression of *PGC-1α* and *NRF-1* to improve mitochondrial function, such as mitochondrial membrane potential, mitochondrial respiration, and ATP synthesis [29].

Noteworthy, Lee et al. found an increase in mitochondrial mass and mtDNA content in a concentration- and dose-dependent manner in human lung fibroblast cell lines treated with H_2_O_2_ at concentrations of 90–360 µM for 24–72 h. These results highlight the role of mtDNA alterations as early molecular events in response to endogenous or exogenous oxidative stress [27]. In another elegant study, both replicative and H_2_O_2_-induced premature senescence models were used to detect the mitochondrial biological characteristics in human embryonic lung fibroblasts. The accumulated mtROS induced cellular senescence, whether replicative or premature, with the common features of low-level mitochondria quantity, mitochondrial 5-methylcytosine (mt-5-mC) content and mitochondrial transcription factor A (*mtTFAM*) protein expression, and high-level mtDNA copy number and DNA methyltransferase (*mtDNMT*) activity being the compensatory effects [30].

## 4. Materials and Methods

### 4.1. H_2_O_2_ Experiments with Peripheral Blood Mononuclear Cells

Human primary PBMCs from anonymous individuals were purchased from Lonza (Walkersville, MD, USA). Cryopreserved PBMCs were thawed in a 37 °C water bath for 1–2 min, centrifuged at 200× *g* at room temperature for 15 min, and suspended in 1 mL of medium LGM-3™, following manufacturer’s instructions (Lonza, Walkersville, MD, USA). Cells were counted in a cell counter (TC20 Automated Cell Counter, Biorad, Hercules, CA, USA), and cell viability was determined via the trypan blue exclusion method. Cell viability was greater than 90%.

A total of ~5 × 10^6^ PBMC were incubated in each 5 mL conical tube with Gibco™ PB-MAX™ Karyotyping Medium (Thermo Fisher Scientific, Waltham, MA, USA) supplemented with fetal bovine serum (FBS), L-glutamine, and phytohemagglutinin (PHA) for subsequent analyses. All cells were cultured in a humidified atmosphere (5% CO_2_, 37 °C).

H_2_O_2_ was applied as an exogenous inducer of oxidative stress. Specifically, PBMCs were treated with 100 μM or 200 μM H_2_O_2_ (Merck, Darmstadt, Germany) after 24 h of incubation. Each test contained three separate experiments: (1) PBMCs without H_2_O_2_ treatment (control); (2) PBMCs treated with 100 μM H_2_O_2_ (mild oxidative stress exposure); (3) PBMCs treated with 200 μM H_2_O_2_ (high oxidative stress exposure).

Afterward, after 44 h, 72 h, and 96 h of incubation, cells were washed twice with DPBS, spun down to a pellet, and suspended in 200 µL DPBS for DNA extraction and subsequent analyses (telomere length and mtDNAcn assays). In parallel, for chromosomal damage analysis, cells were harvested at 72 or 96 h post-PHA stimulation and fixed as described below. To demonstrate reproducibility, two independent experiments were carried out for PBMCs derived from two different healthy individuals. Figure 5 shows the schematic diagram for the experimental setup.

### 4.2. Telomere Length (TL) and Mitochondrial DNA Copy Number (mtDNAcn) Analysis

DNA was extracted from PBMCs by using the QIAamp DNA Mini Kit (Qiagen, Hilden, Germany), according to the manufacturer’s instructions. DNA concentration and quality were assessed using a NanoDrop Lite Spectrophotometer (Thermo Scientific, Waltham, MA, USA). An absorbance ratio at both 260 and 280 nm (A260/A280) greater than 1.7 was considered suitable for the subsequent analyses

Both TL and mtDNAcn levels were measured by using quantitative real-time (RT) methods (CFX384 Touch Real-time PCR detection system, Bio-Rad, Hercules, CA, USA), following previously described protocols. In brief, TL was measured in genomic DNA by determining the T/S ratio. A relative telomere length was calculated using the equation T/S ratio = 2^−ΔΔCt^, where Ct is a threshold cycle and ΔCt = Ct × telomere − Ct × single copy gene. The T/S ratio reflected the average length of the telomeres across all PBMCs [31]. The levels of mtDNAcn were determined through the amplification of the mitochondrial ND1 gene and β-globin gene of genomic DNA. The difference in the average threshold cycle (Ct) number values was used for the measurement of relative content. mtDNAcn was calculated by using the (2^−ΔCt^) method (ΔCt = Ct mtNDI1 − Ct gDNA). All RT PCRs were performed in triplicate in 384-well plates [32].

### 4.3. Micronucleus Assay

The presence of MN in binucleated cells was assayed by blocking the cells at the cytokinesis stage [18]. In brief, the PBMC culture was set as described earlier in the text. Cyt-B (6 µg/mL) was added to the culture at 44 h after the initiation. After a total of 72 h or 96 h of incubation, cells were harvested and fixed according to the standard methods. The fixed cells were dropped onto clean iced slides, air-dried, and stained using the Giemsa technique. For each sample, 1000 binucleated cells were scored under an optical microscope (final magnification 40×) for MN analysis, following the criteria for MN acceptance [18]. We evaluated the MN frequency as the number of micronucleated cells scored per 1000 observed binucleated cells. Figure 6 shows a schematic representation of the MN formation.

### 4.4. Statistical Analysis

The results were expressed as the mean ± standard error (SE). Continuous variables were compared using the Student *t*-test and Mann–Whitney U test for parametric and non-parametric data, respectively. Multiple comparisons were performed via one-way analysis of variance (ANOVA), followed by a multiple comparison test (Bonferroni test), or a Kruskall–Wallis test. Regression analysis with the Pearson test was used to evaluate the relationship between two continuous variables. Statistical analysis of the data was conducted using the StatView statistical package, version 5.0.1 (Abacus Concepts, Berkeley, CA, USA). A *p*-value of ≤0.05 was considered statistically significant in this study. 

## 5. Conclusions

In summary, telomere shortening/dysfunction may be the main triggering mechanism of chromosomal damage in stimulated T-lymphocytes under the stimulation of oxidative stress. An endogenous upregulation of mtDNA content could be an early compensatory mechanism to sustain oxidative phosphorylation activity in response to stress. Moreover, our findings showed a direct link between telomere dysfunction, chromosomal damage, and mtDNA content. Further investigations are needed to understand the molecular mechanisms involved in telomere crosstalk with mtDNA and how these processes can affect the pathophysiology of many age-related diseases. This research is essential for revealing new pathogenic pathways and allowing the development of anti-aging drugs. Numerous anti-aging strategies, grounded in the two primary hallmarks of aging, have thus far been developed. These encompass telomere reactivation, mitophagy activation, the employment of epigenetic drugs, the removal of senescent cells, the administration of anti-oxidant and anti-inflammatory drugs, and the utilization of stem cell-based therapy. For instance, achieving optimal telomere length through the use of telomerase activators or restoring mitochondrial function via SIRT1 and AMPK activation are potentially effective approaches for counteracting aging [33]. Consequently, delving into the molecular intricacies of the aging process undoubtedly represents a fascinating challenge

## Figures and Tables

**Figure 1 ijms-25-01428-f001:**
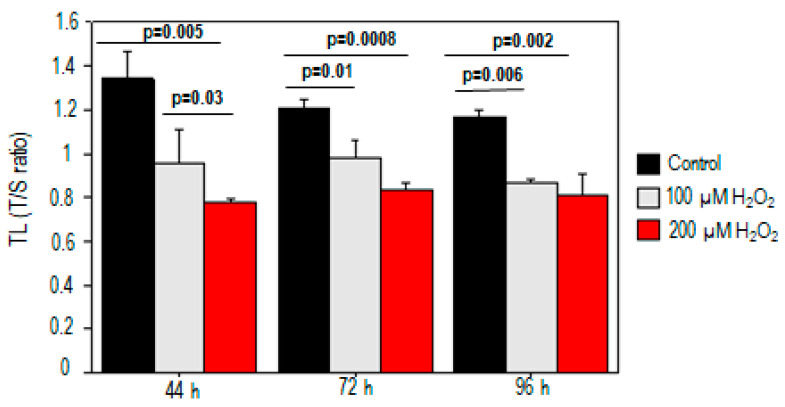
Telomere shortening in PBMCs treated with H_2_O_2_. Telomere shortening was observed after exposure to both doses of H_2_O_2_ compared to the control at different times.

**Figure 2 ijms-25-01428-f002:**
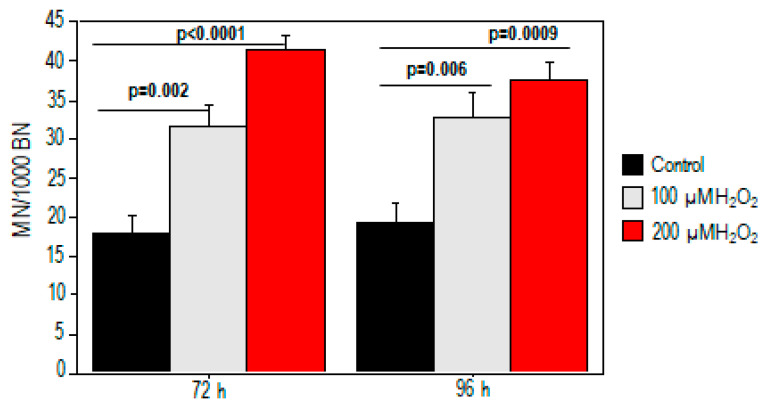
Increased chromosomal damage in PBMCs in response to H_2_O_2_ treatment. Micronucleus yield increased at 72 h and 96 h for both 100 µM and 200 µM H_2_O_2_ exposure.

**Figure 3 ijms-25-01428-f003:**
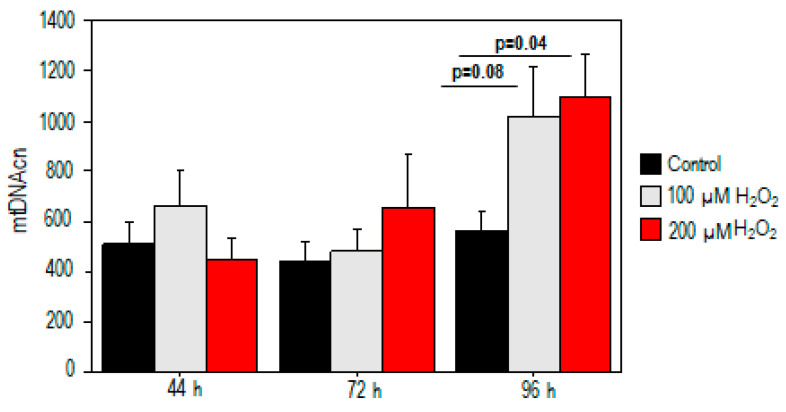
Mitochondrial DNA copy number alterations in PBMCs treated with H_2_O_2_. mtDNAcn increased after high oxidative stress exposure (200 µM H_2_O_2_) compared to the control value.

**Figure 4 ijms-25-01428-f004:**
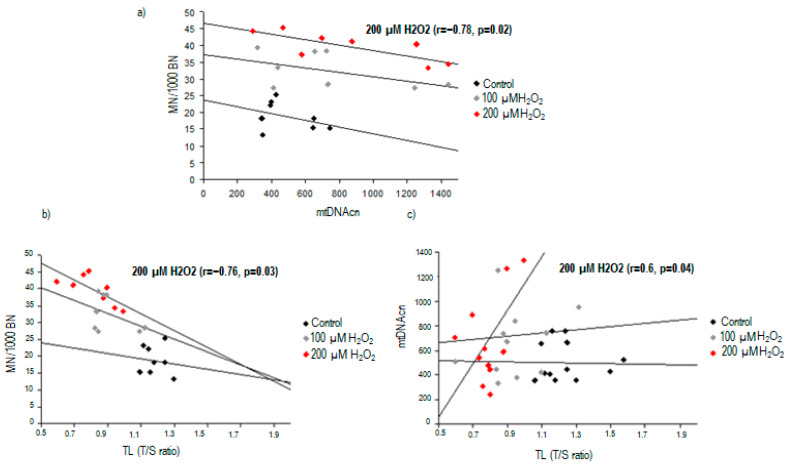
Correlations between nuclear DNA damage and mitochondrial DNA content in response to 200 μM H_2_O_2_ exposure: (**a**) correlation between chromosomal damage and mtDNAcn; (**b**) correlation between chromosomal damage and telomere length; (**c**) correlation between mtDNAcn and telomere length.

**Figure 5 ijms-25-01428-f005:**
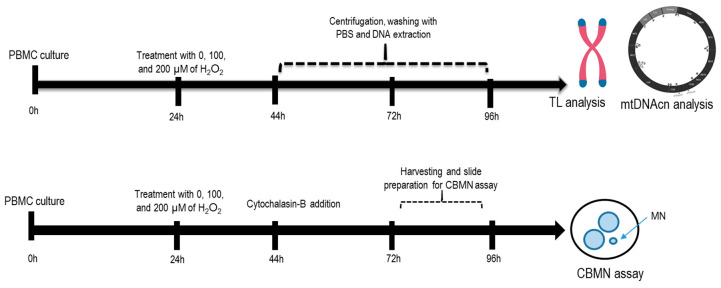
Schematic diagram for the experimental setup. PBMCs were treated with 100 μM or 200 μM H_2_O_2_ at 24 h incubation. At 44 h, 72 h, and 96 h incubation, DNA was extracted from cells for subsequent analyses (telomere length and mtDNAcn assays). For MN analysis, cells were harvested at 72 or 96 h post-PHA stimulation.

**Figure 6 ijms-25-01428-f006:**
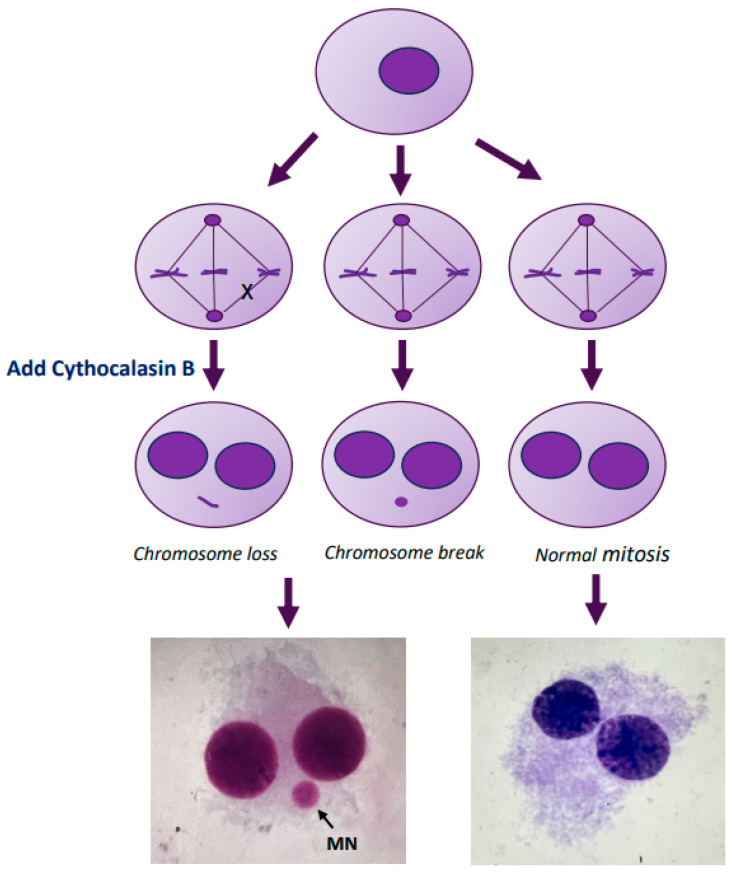
Schematic representation of the micronucleus formation. Micronuclei (MN) are tiny extra-nuclear bodies in the cytoplasm consisting of acentric fragments of chromosomes or entire chromosomes, which do not integrate into the daughter nuclei during cell division.

## Data Availability

The data presented in this study will be made available by the authors on request.

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
