# Peer review of "Telomere Length, Mitochondrial DNA, and Micronucleus Yield in Response to Oxidative Stress in Peripheral Blood Mononuclear Cells"

_ijms, 2024, doi:10.3390/ijms25031428_

Round 1

Reviewer 1 Report

Comments and Suggestions for Authors

In this manuscript, the authors explore the association of Telomere length, mitochondrial DNA and micronucleus yield in response to H2O2 treatment. Firstly, the authors observed a shortening in TL and a concomitant increase in chromosomal DNA damage after treatment with H2O2 in PBMCs. Next, the authors observed a concomitant increase in MN at 72h and persisted at 96h. Moreover, in PBMCs treated with 200μM H2O2, a significant inverse correlation was found between TL and MN, and mtDNA content was directly correlated with TL and inversely related to MN. In general, this manuscript just presents a phenotype induced by H2O2 treatment without any possible mechanism. So I think this manuscript is not suitable for publication in IJMS.

Specific comments:

1. In Figure 1-4, lacking figure legend.

2. In figure 3, please indicate the correct P-value between 100μM H2O2 and 200μM H2O2 treatment.  

3. In this manuscript, the authors did not present any possible mechanism to explain

the phenotype induced by H2O2 treatment.      

Author Response

Thank you for reviewing our manuscript and for your invaluable comments. We have addressed all identified errors and included the missing p-values in the figures. Currently, there is a growing interest in a novel hypothesis regarding aging, highlighting dysfunctional telomeres and mitochondria as key players in a vicious cycle that promotes cellular aging. The intricate relationship between telomere erosion and mitochondrial injury remains poorly understood, raising questions about whether one induces the other, and the potential mechanisms underlying this dual damage. Moreover, there is a significant gap in the existing literature, with no published studies investigating the concurrent effects of ROS generation and oxidative stress on telomere length, chromosomal damage, and alterations in mtDNA content in human peripheral cells. In light of these gaps, our pilot study aimed to provide insights into the biological interplay between telomere shortening, chromosomal instability, and mtDNA dysfunction in response to ROS generation and oxidative stress in cells. We agree with the reviewer's comment that further research is necessary to elucidate the precise mechanisms linking telomere shortening and mitochondrial dysfunction in age-related diseases.

Reviewer 2 Report

Comments and Suggestions for Authors

Borghini et al. ijms-2795768 " Telomere length, mitochondrial DNA, and micronucleus yield in response to oxidative stress in peripheral blood mononuclear cells " is a valuable paper showing telomere shortening is possible to be the main triggering mechanism of chromosomal damage in stimulated T-lymphocytes under the stimulation of oxidative stress. However, some points are difficult for the reviewer to understand. The reviewer hopes that providing more information (described below) will improve the quality of this study. 

Major comment

1. The numbers and letters used in all figures are difficult to read. Therefore, the reviewer believes that this point should be improved to clearly show the results. 

2. The legends are not described in all figures, and the title of the figure is not good. The reviewer believes that the Legends in Figures are important for readers to understand your experiments effectively. The authors have attached the legend to all the figures. 

3. Some p-values are presented outside the panel. The reviewer believes that this value should be included in the panel. Authors should improve this point.

4. Page 2 and 3, Figure 1 and 3 – In Figure 1 and 3, the authors chose 44, 72, and 96 h as the incubation time. In this case, we chose 48 h to avoid selecting 44 h. Why did you select 44 h as the incubation time? 

5. Page 3 and 4, Figure 2 and 4 –the authors used 1000 BN in these figures. I did not understand about 1000 BN. What is 1000 BN? Is it general term? 

6. All experiments were conducted exclusively on H202. The reviewer believes that the authors should conduct similar experiments under other ROS-producing conditions.

Minor comment

1. The authors use both H202 and H2O2 in this manuscript. The reviewer believes that the authors should be unified on one of them. 

2. Author have described 100µM or 72h in this manuscript. However, a space must be inserted between the number and unit.

Author Response

Borghini et al. ijms-2795768 " Telomere length, mitochondrial DNA, and micronucleus yield in response to oxidative stress in peripheral blood mononuclear cells " is a valuable paper showing telomere shortening is possible to be the main triggering mechanism of chromosomal damage in stimulated T-lymphocytes under the stimulation of oxidative stress. However, some points are difficult for the reviewer to understand. The reviewer hopes that providing more information (described below) will improve the quality of this study.

Thanks for your kind comments.

Major comment

  1. The numbers and letters used in all figures are difficult to read. Therefore, the reviewer believes that this point should be improved to clearly show the results.
  2. The legends are not described in all figures, and the title of the figure is not good. The reviewer believes that the Legends in Figures are important for readers to understand your experiments effectively. The authors have attached the legend to all the figures.

As suggested by the reviewer, we better described the figures by adding the legends.

  1. Some p-values are presented outside the panel. The reviewer believes that this value should be included in the panel. Authors should improve this point.

Accordingly, we have made the requested changes.

  1. Page 2 and 3, Figure 1 and 3 – In Figure 1 and 3, the authors chose 44, 72, and 96 h as the incubation time. In this case, we chose 48 h to avoid selecting 44 h. Why did you select 44 h as the incubation time?

We selected a 44-hour incubation time, aligning with the point at which cytochalasin is introduced for the micronucleus assay. Consequently, we maintained this incubation duration for concurrent DNA extraction and subsequent experiments, including telomere length and mtDNAcn analysis.

  1. Page 3 and 4, Figure 2 and 4 –the authors used 1000 BN in these figures. I did not understand about 1000 BN. What is 1000 BN? Is it general term?

We evaluated the MN frequency as the number of micronucleated cells scored per 1,000 observed binucleated cells and reported the results as MN/1000 BN. However, we have provided a better clarification in the Methods Section.

  1. All experiments were conducted exclusively on H202. The reviewer believes that the authors should conduct similar experiments under other ROS-producing conditions.

Hydrogen peroxide (H2O2) is one of the most potent available oxidizing agents and plays a crucial role as a metabolite in numerous redox metabolism reactions and various physiological processes within cells. H2O2 is a widely recognized source of oxidative damage in genetic toxicology, influencing biological responses and causing DNA damage in cells. We agree with the reviewer's comment that further research, also under other ROS-producing conditions, is needed to elucidate the molecular mechanisms linking telomere shortening and mitochondrial dysfunction.

Minor comment

  1. The authors use both H202 and H2O2 in this manuscript. The reviewer believes that the authors should be unified on one of them.
  2. Author have described 100µM or 72h in this manuscript. However, a space must be inserted between the number and unit.

 We have modified the text as suggested by the Reviewer.

Reviewer 3 Report

Comments and Suggestions for Authors

A well written and pithy study, highlighting the the effects on Telomere length, mitochondrial DNA, and micronucleus yield in response to oxidative stress in peripheral blood mononuclear cells. The appropriate experimental design and techniques are employed. The data is well interrogated, interpreted and presented. The conclusions drawn are relevant. I would like to see the authors speculate further on the potential molecular mechanisms linking the observed results. Are the observations described in the study correlation or causation in this instance, and how do the draw such conclusions.

Points

I would like to see representative images go the MN assay and analysis, for context.

In the discussion, can the authors speculate or outline the potential molecular  mechanisms linking the main theories of cellular aging i.e. specifically  mitochondria dysfunction and telomeres attrition.

The final statement in the discussion is valuable and worth expanding/discussing more-

'low-level mitochondria quantity, mitochondrial 5-methylcytosine (mt-5-mC) content and mitochondrial transcription factor A (mtTFAM) protein expression, and high-level mtDNA copy number and DNA methyltransferase ( mtDNMT) activity as the compensatory effects.'- Do the authors have supporting data from their experimental set up to support this hypothesis and paradigm?

Materials and Methods

Ln235- please list reference(s) for these protocols.

Author Response

A well written and pithy study, highlighting the effects on telomere length, mitochondrial DNA, and micronucleus yield in response to oxidative stress in peripheral blood mononuclear cells. The appropriate experimental design and techniques are employed. The data is well interrogated, interpreted and presented. The conclusions drawn are relevant. I would like to see the authors speculate further on the potential molecular mechanisms linking the observed results. Are the observations described in the study correlation or causation in this instance, and how do the draw such conclusions.

Thank you very much for your kind words about our manuscript.

Points

I would like to see representative images go the MN assay and analysis, for context.

Accordingly, we added a new Figure (Figure 6) showing a schematic representation of the micronucleus formation.

In the discussion, can the authors speculate or outline the potential molecular mechanisms linking the main theories of cellular aging i.e. specifically  mitochondria dysfunction and telomeres attrition.

We better described the main potential molecular mechanism linking the main theories of cellular aging (mitochondrial dysfunction and telomere attrition).

The evidence suggests that mitochondrial DNA (mtDNA) damage plays a role in contributing to mitochondrial dysfunction and subsequent reactive oxygen species (ROS) production, thereby intensifying telomere damage in a cyclic manner [7,9]. It is noteworthy, however, that mitochondria can also become dysfunctional as telomeres shorten, establishing a connection between the primary theories of cellular aging [25]. Telomere dysfunction triggers the suppression of the master regulator of mitochondrial biogenesis and function, namely, peroxisome proliferator-activated receptor gamma co-activator 1α/β (PGC-1α/β), resulting in mitochondrial dysfunction and ROS production. Specifically, telomere attrition activates p53 and DNA damage response pathways, which in turn inhibit PGC-1α and PGC-1β. Consequently, mitochondrial dysfunction driven by telomere shortening may impair oxidative phosphorylation and increase ROS generation. This can further exacerbate mitochondrial distress and telomere shortening, creating a two-pronged vicious cycle that amplifies the dysfunctional system [26].

The final statement in the discussion is valuable and worth expanding/discussing more.

As suggested, we discussed our final statement as follows:

Numerous anti-aging strategies, grounded in the two primary hallmarks of aging, have been developed thus far. These encompass telomere reactivation, mitophagy activation, employment of epigenetic drugs, removal of senescent cells, administration of antioxidant and anti-inflammatory drugs, and the utilization of stem cell-based therapy. For instance, achieving optimal telomere length through the use of telomerase activators or restoring mitochondrial function via SIRT1 and AMPK activation stands out as potentially effective approaches to counteract aging [33]. Consequently, delving into the molecular intricacies of the aging process undoubtedly represents a fascinating challenge.”

'low-level mitochondria quantity, mitochondrial 5-methylcytosine (mt-5-mC) content and mitochondrial transcription factor A (mtTFAM) protein expression, and high-level mtDNA copy number and DNA methyltransferase (mtDNMT) activity as the compensatory effects.'- Do the authors have supporting data from their experimental set up to support this hypothesis and paradigm?

Regrettably, we lack experimental data to substantiate this hypothesis. Our study serves as a pilot investigation designed to offer insights into the biological interplay among telomere shortening, chromosomal instability, and mitochondrial DNA dysfunction in response to H2O2 in cells. Additional research is necessary to unravel the intriguing molecular mechanisms connecting telomere shortening and mitochondrial dysfunction.

Materials and Methods

Ln235- please list reference(s) for these protocols.

We added a sentence in the manuscript as the protocol refers to the manufacture’s instructions.

Round 2

Reviewer 1 Report

Comments and Suggestions for Authors

Accept

Author Response

Thanks for your revision.

Reviewer 2 Report

Comments and Suggestions for Authors

The second revised paper seems to be improved and should be worth publishing in this journal. However, the reviewer hopes that change one piece of information (described below).

Figure. 1-5– The resolution of the text on the vertical and horizontal axes of these figures is low. The reviewer recommends increasing the resolution of these as they are difficult to read.

Author Response

The second revised paper seems to be improved and should be worth publishing in this journal. However, the reviewer hopes that change one piece of information (described below).

Figure. 1-5– The resolution of the text on the vertical and horizontal axes of these figures is low. The reviewer recommends increasing the resolution of these as they are difficult to read.

Thank you for your kind words about our revised manuscript. Accordingly, we modified the resolution of the text on the vertical and horizontal axes of the figures.